# Antifungal Thiazolidines: Synthesis and Biological Evaluation of Mycosidine Congeners

**DOI:** 10.3390/ph15050563

**Published:** 2022-05-01

**Authors:** Igor B. Levshin, Alexander Y. Simonov, Sergey N. Lavrenov, Alexey A. Panov, Natalia E. Grammatikova, Alexander A. Alexandrov, Eslam S. M. O. Ghazy, Nikita A. Savin, Peter V. Gorelkin, Alexander S. Erofeev, Vladimir I. Polshakov

**Affiliations:** 1Gause Institute of New Antibiotics, 11 B. Pirogovskaya Street, 119021 Moscow, Russia; levshin@panavir.ru (I.B.L.); simonov-live@inbox.ru (A.Y.S.); satory@mail.ru (S.N.L.); ngrammatikova@yandex.ru (N.E.G.); 2Bach Institute of Biochemistry, Federal Research Center of Biotechnology of the RAS, 119071 Moscow, Russia; alexvir@inbi.ras.ru (A.A.A.); 1072195050@rudn.ru (E.S.M.O.G.); 3Institute of Biochemical Technology and Nanotechnology, Peoples’ Friendship University of Russia (RUDN), 6 Miklukho-Maklaya Street, 117198 Moscow, Russia; 4Department of Microbiology, Faculty of Pharmacy, Tanta University, Tanta 31111, Egypt; 5Research Laboratory of Biophysics, National University of Science and Technology “MISiS”, 4 Leninsky Ave., 119049 Moscow, Russia; nsavin99@mail.ru (N.A.S.); peter.gorelkin@gmail.com (P.V.G.); erofeev@polly.phys.msu.ru (A.S.E.); 6Faculty of Fundamental Medicine, Lomonosov Moscow State University, 27/1 Lomonosovsky Ave., 119991 Moscow, Russia; vpolsha@fbm.msu.ru

**Keywords:** thiazolidine-2,4-dione, antifungal activity, drug design, glucose transport, cell wall

## Abstract

Novel derivatives of Mycosidine (3,5-substituted thiazolidine-2,4-diones) are synthesized by Knoevenagel condensation and reactions of thiazolidines with chloroformates or halo-acetic acid esters. Furthermore, 5-Arylidene-2,4-thiazolidinediones and their 2-thioxo analogs containing halogen and hydroxy groups or di(benzyloxy) substituents in 5-benzylidene moiety are tested for antifungal activity in vitro. Some of the synthesized compounds exhibit high antifungal activity, both fungistatic and fungicidal, and lead to morphological changes in the *Candida* yeast cell wall. Based on the use of limited proteomic screening and toxicity analysis in mutants, we show that Mycosidine activity is associated with glucose transport. This suggests that this first-in-class antifungal drug has a novel mechanism of action that deserves further study.

## 1. Introduction

A limited number of chemical classes of compounds can be used against fungal pathogens at this time. In particular, the treatment of topical and systemic fungal diseases in humans and animals uses drugs belonging to the following classes: azoles, polyenes, nucleobase analogues, and echinocandins. Azole drugs inhibit the synthesis of ergosterol, which is a component of lipid membranes, via inactivation of the intracellular enzyme lanosterol demethylase. Polyenes bind to ergosterol on the cell surface, thereby disrupting the membrane structure. Flucytosine inhibits DNA and possibly RNA synthesis. Echinocandins disrupt the structure of the fungal cell wall by inhibiting 1,3-beta-glucan synthase [1,2,3]. Thus, these drugs target only a few molecular targets, which is why the problem of resistant forms of fungal pathogens arises more and more often in recent years. In addition, some of these drugs have significant side effects. Notably, during the past 30 years, only one new antifungal drug—Ibrexafungerp, selectively acting on the cell wall of pathogenic fungi, has entered clinical practice [4].

The search for new antifungal drugs and new targets is an extremely urgent task. One of the promising directions in the search for new antifungal drugs is the detection of chemical compounds acting on the cell wall of the fungus [5,6,7,8,9].

Although the exact structure of the fungal cell wall is not fully understood, it consists of a complex mixture of proteins and polysaccharides, including glucan, mannans, and chitin. Most of the major cell wall components of fungal pathogens are not represented in humans. For these reasons, enzymes that assemble components of the fungal cell wall are excellent targets for antifungal chemotherapies and fungicides [10,11].

At the beginning of the 21st century, the first inhibitors of the mannosyl transferase—derivatives of 2-(4-oxo-2-thioxothiazolidin-3-yl)acetic acid—were described (Figure 1, structure I) [12,13].

It was shown [12,13] that transposition of the benzyloxy group to the adjacent 3-position on the central aromatic ring had little effect on the PMT1 inhibitory activity. However, 3,4-bis(benzyloxy) substitution resulted in a significant increase in activity, both against the enzyme and *C. albicans* cells (IC_50_: 2.3 and 3.01 µM). Several compounds have been identified which inhibit *C. albicans* PMT1 with an IC_50_ in the range 0.2–0.5 µM (R=H, Me) [13].

Thiazolidine and its derivatives, such as thiazolidine-2,4-dione (TZD) (Figure 1, structure **II**), are important heterocyclic systems with a range of biological activities [14,15,16,17,18,19,20,21,22,23,24,25,26,27]. According to the literature, TZD is one of the most important derivatives, due to a wide range of therapeutic effects and the possibility of modifications producing additional biological activities. These include antiarthritic [28], antimicrobial [29,30,31], antifungal [32], anti-inflammatory [33,34], anticonvulsant, and antioxidant [35] activity, among others. The ability of TZDs to contribute to cancer therapy has been shown in numerous in vitro and in vivo studies [36]. 

The initial impulse for intensive study of the chemistry and biology of thiazolidines was initiated by the launch of hypoglycemic drugs—glitazones—containing various substituents in the 5-benzyl substituent of the thiazolidine core (Figure 2) [37]. While these TZD derivatives are known to stimulate the PPAR-γ receptor, they also have multiple PPAR-γ independent effects and the specific role of PPAR-γ activation in the anticancer effects of TZDs is still under investigation [38,39].

TZDs have exhibited anti-tumor activity in a wide variety of experimental cancer models via effects on the cell cycle and induction of cell differentiation and apoptosis as well as by inhibiting tumor angiogenesis [40,41]. A study reported by Shah [40] showed that TZD derivative ciglitazone significantly decreased the production of VEGF in human granulosa cells in an in vitro model [42]. Mechanisms of anticancer activity of thiazolidin-4-ones and related heterocycles may be associated with their affinity to anticancer biological targets, such as non-membrane protein tyrosine phosphatase (SHP2), JNK-stimulating phosphatase-1 (JSP-1), tumor necrosis factor TNF-α, antiapoptotic biocomplex Bcl-XL-BH3, and integrin αvβ3 [43].

Epalrestat, a derivative of TZD, is also used in medical practice as a non-competitive and reversible aldose reductase inhibitor for the treatment of diabetic neuropathy [44] (Figure 3).

Our group is interested in the antimicrobial activity of thiazolidine-2,4-diones, and in particular, their antifungal effects. A systematic study of a series of various 3- and 5-substituted thiazolidines leads to the creation of a novel topical antifungal drug—Mycosidine (Figure 3) [45,46,47,48].

Mycosidine^®^, chemically known as *(Z)*-5-(4-chlorobenzylidene)-3-methoxycarbonyl-1,3-thiazolidine-2,4-dione, was approved by the Russian Ministry of Health in 2008 for the topical treatment of interdigital tinea pedis, tinea cruris, and tinea corporis caused by dermatophytes (*Microsporum canis*, *Trichophyton gyrseum*, *Trichophyton mentagrophytes*, *Trichophyton rubrum*) and yeasts (*Candida* spp.). This compound belongs to a new class of topical antifungal drugs with an uncharacterized mechanism of action; however, it was shown to cause disruption of the fungal cell wall [49].

Thus, this study aims to create novel, more efficient derivatives of TZD and to characterize the mechanisms responsible for their antifungal effects.

## 2. Results

### 2.1. Ethyl 2,4-Dioxothiazolidine-3-Carboxylate and Other Thiazolidine-3-Carboxylates

The interactions of chloroformates with TZD derivatives containing hydroxyl groups in the arylidene fragment have recently been described in the literature [50,51]. The reactions were carried out in dry acetone in the presence of potassium carbonate. With an equimolecular ratio of reagents, the substitution on only the phenolic hydroxyl group is described [50], and with the double equivalent of the chloroformate and potassium carbonate, two reaction centers were involved at once with substitution at the ring nitrogen atom and the hydroxyl group of the arylidene radical [51]. The only *N*-ethoxycarbonyl derivative, obtained from the potassium salt of 5-nitrofurfurilidene-thiazolidine-2,4-dione with ethyl chloroformate in DMF, was described in a French patent [52]. However, other than the melting point, no data confirming the structure of the products was given.

We developed another method for obtaining new similar alkoxycarbonyl derivatives of TZD, on which the synthesis of Mycosidine is based: namely, reacting 5-arylidene TZD in a nonpolar solvent (e.g., toluene) with methyl chloroformate and triethylamine as a base at room temperature [45]. Using this technique from TZD (**1**) and ethyl chloroformate, ethyl 2,4-dioxothiazolidine-3-carboxylate (**3**) unsubstituted at position five was obtained.

The structure of compound **3** was confirmed using NMR spectroscopy and mass spectrometry. In the ^1^H NMR spectrum of **3**, there is a singlet of the methylene group at position five of the ring in the region of 4.00 ppm and signals of the ethyl group in the region of 4.38 ppm (2H, q, *J* = 7.16) and 1.34 ppm (3H, t, *J* = 6.82). In the ^13^C NMR spectrum (in CDCl_3_), signals of three carbonyl groups are observed at 147.49 (COOH), 167.55 (C^4^=O), and 167.65 (C^2^=O) ppm, the methylene group of the thiazolidine ring at 65.88 ppm, and the signals of the ethyl group of the ester CO-CH_2_CH_3_ at δ 33.93 (CH_2_) ppm and 13.81 (CH_3_) ppm.

However, it was not possible to carry out the Knoevenagel reaction of **3** with aromatic aldehydes by direct condensation under typical conditions. Under milder conditions (heating in ethanol with piperazine), decomposition products and unreacted compounds were isolated. In acetic acid with the catalysts (methylamine, piperidine), upon heating, the alkoxycarbonyl group was completely removed with the release of TZD as the main product. We managed to obtain the target 5-arylidene-3-carboxylates using an alternative method starting from 5-(4-chlorobenzylidene)thiazolidine-2,4-dione (**4a**) with various chloroformates (Figure 1).

Numerous ways of obtaining 5-arylidene derivatives of TZD using the Knoevenagel reaction are described in the literature. Various conditions are described, e.g., for base catalysts, such as KOH or NaOH: Et_3_N, AcONa or AcONH_4_ in AcOH, or toluene at 110 °C with addition of MeNH_2_; in AcOH, ethanol or toluene with piperidine; and in the presence of AcOH, PEG-400, as well as pyrrolidine and morpholine [14,15,16,17,18,19,20,21,22]. 

Compound **4a** was obtained by the method described by us earlier [53] from compound **1** and 4-chlorobenzaldehyde in the acetic acid medium with 33% aqueous solution of methylamine as a catalyst. Condensation of **4a** with various chloroformates under conditions similar to the preparation of **3** proceeded successfully with the production of target crystalline derivatives with good yields (68–85%). Structures were confirmed by 1D and 2D NMR and HRMS.

An example of the structure of **5a** as a prototype of a series of compounds **5a**–**e** is shown in Figure 4. The values of chemical shifts of ^1^H and ^13^C are shown in Table 1 and Table 2, respectively.

The carbonyl group C(4) signal in the ^13^C monoresonance spectrum is observed as a doublet with a coupling constant of 6.9 Hz. This information unambiguously confirms the attribution of this signal, because splitting on the proton H(6) can be observed only for C(4) but not for C(2). This measurement also clearly indicates the *Z*-configuration of the molecule, in which the nuclei C(4) and H(6), separated by three chemical bonds, are in a *cis* position to each other. For an alternative E-configuration in which these nuclei are in a *trans* position to each other, the spin-spin interaction constant between them should reach a value of 13–15 Hz. Thus, **5a**, like the entire **5a**–**e** series, exists exclusively in the form of a *Z*-conformer.

The description of NMR and mass spectra confirming the structure of the remaining alkoxycarbonyl derivatives **5a**–**e** is given in the experimental section. In the IR spectra, there are three intense bands of valence vibrations of the CO group in the region: 1784–1789 (COOR), 1742–1747 (C^4^=O), and 1690–1700 cm^−1^ (C^2^=O). Valence vibrations are CH=C and the aromatic ring is in the region of 1606 and 1586 cm^−1^.

### 2.2. 5-Arylidene-3-Benzoylthiazolidine-2,4-Dione Derivatives

Several articles have described 3-Benzoyl derivatives of 5-arylidene-thiazolidine-2,4-dione as potential herbicidal and antimicrobial agents. Their synthesis was based on 5-arylidene-thiazolidine-2,4-dione and the corresponding benzoyl chloride and it had a good yield in anhydrous acetone in the presence of K_2_CO_3_ with heating [54] or in pyridine at 70 °C [55].

A more detailed study of the benzoylation reaction of TZD unsubstituted at position 5 was carried out by us earlier [56]. It was found that, depending on the synthesis conditions, there is a possibility of the formation of a 3-benzoyl derivative of TZD (**6a**) and thiazole-2,4-diyldibenzoate (**6b**) in addition to the main reaction product (Figure 2).

In this work, we tested the possibility of condensation of 3-benzoyl-TZD (6a) with aromatic aldehydes using the Knoevenagel reaction. The goal in this case was to obtain compounds with an unsubstituted hydroxyl group in the benzylidene fragment, because the acylation of the phenolic group of the arylidene residue may cause formation of unwanted *O*-acylated byproducts [57]. However, we were not able to obtain 5-arylidene derivatives **7a**,**b** by direct condensation of **6a** with aromatic aldehydes under any conditions. Then, a well-known alternative method for obtaining **7a**,**b** by benzoylation of 5-arylidene TZD **4a**,**b** was used (Figure 3). In addition, some derivatives of **7a**,**b** were obtained that did not contain free hydroxyl groups in the phenyl ring. The reaction was carried out according to a published procedure using acetone with K_2_CO_3_ under heating [54].

When analyzing the structure of 3-benzoyl derivatives by NMR (**6a**, **7a**), it was noticed that upon storage in a DMSO-*d*_6_ solution, the benzoyl group undergoes decomposition with the formation of benzoic acid. This process occurs for both the unsubstituted TZD **6a** (see Appendix A) and the 5-arylidene derivative **7a** (see Appendix A). With no substituents in position 5 of the thiazolidinedione ring, the 3-benzoyl derivative TZD **6a** hydrolyses in a solution of DMSO-*d*_6_ with a half-life of 5 h at 25 °C. The half-life of 5-substituted 3-benzoyl-TZD **7a** in identical conditions was 11 h. A similar hydrolysis reaction also occurs in other solvents containing water. In the methanol solution, more complex reactions, involving the solvent, can occur (see Appendix A).

Stability is a highly important parameter for the active substance of a perspective drug. Thus, due to the instability of 3-benzoyl-substituted derivatives of TZD, we did not continue the study of this series of compounds containing a benzoyl group at the nitrogen atom of the thiazolidine ring.

### 2.3. 2-(2,4-Dioxothiazolidin-3-yl)Acetic Acid and Its Derivatives

Being derivatives of thiazolidines, 2-(2,4-Dioxothiazolidin-3-yl)acetic acids and their thioxo analogs are widely represented in the literature. Most of the works on the synthesis of these compounds are devoted to the search for new hypoglycemic drugs—aldose reductase inhibitors [58,59,60,61,62,63]—or anticancer [64] and antimicrobial drugs [65]. Recently, an article was published showing that several TZD derivatives specifically inhibit the initiation of hyphal growth in *C. albicans* without affecting cell viability or budded growth [66].

To obtain acetic acid derivatives, we used well-known methods of synthesis, with slight changes and optimizations at some stages. We also prepared dioxothiazolidine derivatives and a new 2-thioxo-4-thiazolidinone analogue, previously described in the literature, for comparative activity studies. Both dioxothiazolidine and 2-thioxo-4-thiazolidinone derivatives were prepared and assayed to fully investigate this series.

Synthesis of the target 2-(2,4-dioxothiazolidin-3-yl)acetic acid derivatives **12a**–**e** and their 2-thioxo analogs **17a**,**e** is possible via two paths: either through 5-arylidene derivatives of TZD **4a**–**e** or through 3-substituted TZD **9** or its 2-thioxo derivative **15**. Condensation of commercially available **1**, **14**, and **15** with aromatic aldehydes was carried out according to a previously developed method in boiling acetic acid with a 33% methylamine as a catalyst (method A) [56]. Some of the 5-arylidene derivatives **4**, **10**, and **12** were obtained and described before: **4a [53,67,68,69,70,71,72]**; **4c [69,70]**; **4d [70]**; **10a [57]**; **12b, 17b [12]**). The alkylation reaction for the compounds **4a**–**e** and **15b** with chloroacetic or bromoacetic acid esters was carried out in a DMF solution in the presence of K_2_CO_3_. Then, acidic hydrolysis of the ester group of the thioxo derivative **16** and dioxo derivatives **10**–**11a,b** was performed. The action of a mixture of acetic and hydrochloric acids upon heating led to partial hydrolysis of benzyloxy groups and low yield of **12b**. Under mild conditions of *tert*-butyl ester hydrolysis by trifluoroacetic acid in methylene chloride at room temperature, benzyloxy groups are not affected and thereby the yield of **12b** increases. According to a published method [12], refluxing 2-(2,4-dioxo-1,3-thiazolidin-3-yl)acetic acid with 3,4-di(benzyloxy)benzaldehyde in an acetic acid with sodium acetate gave the product **12b** in a very small yield.

When changing the synthesis conditions (Figure 4) to using *tert*-butyl esters **11b** (obtained by a Knoevenagel reaction with 3,4-dibenzyloxybenzaldehyde) instead of strong acid hydrolysis of ethyl esters, **10b**, by refluxing in HCl/AcOH (method C), and after mild acidic hydrolysis by trifluoroacetic acid at room temperature (method D), the yield of (5-(3,4-bis(benzyloxy)benzylidene)-2,4-dioxothiazolidin-3-yl)acetic acid (**12b**) increased dramatically from 5% [12] to 58%.

Condensation of **9** and 3,4-di(benzyloxy)benzaldehyde at different Knoevenagel conditions in ethanol with piperidine as a catalyst (method B), instead of acetic acid, with CH_3_COONa [12] increased the yield of **12b** only to 14% after 28 h of reflux.

Sulfur-containing counterparts **17b**,**e** were obtained in a similar manner (Figure 5). The compounds containing a hydroxyl group with two chlorine atoms on the phenyl ring (**12d**,**e** and **17e**) were obtained by Knoevenagel condensation of (2,4-dioxothiazolidin-3-yl)acetic acid or commercially available **13** and **14** with corresponding aromatic aldehyde under conditions previously described for other derivatives of TZD: acetic acid as a solvent with 33% methylamine as a catalyst.

In the ^1^H NMR spectra of compounds **4a**–**e**, **10**–**12a**–**e**, **16b**, and **17b**,**e**, the singlet of the CH=C proton of the benzylidene fragment is observed at 7.8–8.0 ppm. The proton of the imide nitrogen of compounds **4a**–**e** is observed in the range from 11 to 12.40 ppm, depending on the arylidene radical. The protons of the methylene group of acetic acid derivatives are observed at 4.52–4.31 ppm.

The ^13^C spectra of all target compounds of groups **4**, **12**, and **14** contain signals of carbonyl groups of the thiazolidine ring at two and four positions located at 168 and 167 ppm and carbonyl of acetic acid at 165 ppm. In the spectra of 2-thioxothiazolidin-4-ones **15**–**17**, the peak of C^2^=S is shifted to the weaker field region compared to the carbonyl group in position four and is observed at 195–193 ppm. The signals of C=O of the benzoyl carbonyl group in compounds **7a**,**b** are at 164 ppm.

During the acquisition of the ^1^H NMR spectra in DMSO*-d6* and CDCl_3_ solutions, a partial decomposition of compounds containing bis-benzyloxy substituents in the phenyl ring was observed, both for **12b** and **4b**. For the thioxo-containing analog **15b**, the hydrolysis process begins in 10–15 min after dissolution in CDCl_3_ (see Appendix A: the process of partial decomposition of **15b** in CDCl_3_).

### 2.4. Evaluation of the Biological Activity of Synthesized Compounds

In vitro minimum inhibitory concentration (MIC) of all target compounds was determined using the method recommended by the National Committee for Clinical Laboratory Standards (NCCLS) Clinical and Laboratory Standards Institute (CLSI) and the serial dilution method in 96-well plates [73,74]. All of the target compounds were evaluated for their antifungal activity against five important fungal pathogens. Fluconazole (FLC) was used as a reference drug. The in vitro antifungal activity results are summarized in Table 3.

As shown in Table 3, the substitution of the nitrogen atom of the thiazolidine ring with an alkoxycarbonyl or benzoyl radical led to a significant increase in antifungal activity compared to unsubstituted derivatives of TZD **4a**–**e**, with respect to all types of yeast and filamentous fungi. The substitution of alkyl groups of alkoxycarbonyl derivatives with phenyl (**5d**) or benzyl (**5e**) radicals negatively affected antimicrobial activity. A similar result was observed with substituted acetic acid esters **10**–**11a,b**.

Dibenzyloxy derivatives of TZD **4b** and its thioxo analog **15b** did not show high antifungal activity. Furthermore, the 2,4-dichloro derivative **12c** exhibited high activity against *A. fumigatus* ATCC 46645 (2 mg/L) and moderate activity against other types of filamentous fungi (16–32 mg/L). 

Other acetic acid derivatives **12d**–**e** containing chloro and hydroxy groups in the benzylidene radical showed moderate antifungal activity against filamentous fungi (MIC = 16 µg/mL) and the 2-thioxothiazolidin-4-one derivative **17e** showed higher activity (MIC = 2–8 mg/L) against *M. canis* B-200 and *T. rubrum* 2002 than the oxy analogue **12e** (MIC = 16 mg/L).

The bis-benzyloxy derivative of TZD **12b** and its thioxo analog **17b** did not show good MIC values, despite the high inhibitory activity of **17b** toward mannosyltransferase reported in some articles [12,13]. This fact indicates that the effect of thiazolidine derivatives on the fungal cell wall may not be associated with the inhibition of PMT1; it also extends to other targets that have yet to be clarified. When the 2-hydroxy group was combined with 3,5-dichloro substituents, the antifungal activity became more pronounced on *C. albicans* yeasts, so an additional experiment on clinical isolates of *Candida* spp. was conducted.

On clinical strains of *Candida* spp. in comparison with Fluconazole and Mycosidine, 3-substituted TZD derivatives **5a** and **7a** showed moderate activity similar to Mycosidine with MIC ranging from 8 to 64 mg/L (Table 4). High activity of 2-hydroxy-3,5-dichloro derivative **12e** was demonstrated on yeasts *C. albicans* ATCC 24433 and *C. parapsilosis* ATCC 22019 (MIC 0.125–0.5 mg/L), whereas activity on other *Candida* spp. was much lower.

### 2.5. Cellular Response to Mycosidine

In order to gain a deeper understanding of the effects of TZD derivatives on fungal cells, we assayed the response of a model fungal organism—*Saccharomyces cerevisiae*—to Mycosidine, which was available in the highest quantities. To do this, we initially determined the MIC (15.6 mg/L) and assayed the number of cells with permeabilized membranes using propidium iodide (PI). Notably, at MIC and below it, the population of dead cells was ~20% after 6 h of treatment, which suggests that the breakdown of the cell wall is not the primary mechanism that stops cell division. We also tested whether Mycosidine had a fungicidal effect on the cells by semi-quantitatively assaying the number of CFU using a spotting assay. This test showed that Mycosidine had a noticeable fungicidal effect, but only after 24 h, with no massive cell death observed after 2, 4, or 6 h of incubation, i.e., cell death in response to the drug is rather slow (Figure 5A).

Then, we tested whether cells treated with this drug at sublethal concentrations experience some specific changes of protein levels. This was done by monitoring the levels of a select number of proteins tagged with GFP using flow cytometry. The proteins we tested were mainly selected from those considered as “sentinel-proteins” [75] as well as several additional proteins (65 proteins and autofluorescence control—see Appendix A). We used different concentrations of the drug—MIC and 0.5×MIC for screening all of the proteins—and then at 0.5–4×MIC for the confirmation tests. Notably, all of the experiments were performed with simultaneous staining for cells with membranes, and the presented data are for living cells only. 

Our results show that Mycosidine caused specific increases in the levels of at least two proteins—Pdr5 and Hxt3 (Figure 5B)—when compared to multiple other drugs that were tested in a similar manner (manuscript in preparation). Notably, this increase was only observed at 0.5×MIC and disappeared at higher concentrations. Pdr5 is an ABC-transporter, involved in the efflux of multiple compounds, including antifungal drugs such as azoles. Hxt3 is a low-affinity glucose transporter, which has not, to our knowledge, been previously implicated in responses to antifungal drugs.

Having identified that these proteins are induced by drug treatment, we tested whether these proteins played a role in the sensitivity of cells to this drug by using strains containing deletions of the corresponding genes [76]. Deletion of Pdr5 had no effect (Figure 5C), which shows that Mycosidine might not be a substrate of the Pdr5 transporter and that its induction might be a non-specific response of the cell to stress caused by the drug. A similar effect was recently shown for the protonophore pentachlorophenol [77].

Interestingly, the deletion of Hxt3 increased the sensitivity of the yeast to the drug (Figure 5C), which suggests that the response of glucose transporters may be necessary for adaptation to the drug. Because yeast has numerous glucose transporters, it might be that one or several of them are targets for Mycosidine or that glucose transport is a resistance mechanism; this requires further study.

Lastly, monitoring of the amount of the GFP-tagged histone Htb2 allows us to monitor the dynamics of the cell cycle of drug treated cells (Figure 5D). This analysis shows that Mycosidine at 0.5×MIC reduces the share of cells with a 2C complement of DNA (because Htb2 levels and DNA levels are usually well correlated) and increases the height of the area between the 1C and 2C peaks, suggesting that progression of the S-phase of the cell cycle is impaired. The MIC concentration of Mycosidine causes most cells to exhibit a 1C distribution, while also causing emergence of an additional population of cells with lower fluorescence, which might suggest apoptotic cell death. This suggests impaired cell cycle progression prior to the initiation of replication as well as, possibly, apoptosis-like cell death.

### 2.6. Effects on the Cell Wall

SICM is a non-contact type of scanning probe microscopy applicable for the investigation of live biological samples in liquid [78,79,80,81]. This method was used to study the influence of antifungal thiazolidines on the topography of living *Candida* cells. It is reasonable to use effective concentrations to see any changes in topography induced by the drugs in question and for this reason concentration equal to MIC and 10×MIC for both Mycosidine and **17b** were chosen. The MIC for Mycosidine is two times lower than the same one for **17b**, but **17b** was still used as it is reported to be a highly active inhibitor of mannosyltransferase. Spikes that were 100–200 nm in diameter were observed after treatment of the *Candida* cells with Mycosidine and **17b** compounds. The appearance of such spikes on the *Candida* cell surface is most probably related to the disruption of the cell wall. Only cells attached to the substrate can be investigated with SCIM and probably for this reason we can easily find some resistant cells without any spikes. The obtained results clearly demonstrate that the action of both Mycosidine and **17b** causes disruption of the cell wall but Mycosidine achieves the same effect with much lower concentration (Figure 6).

## 3. Materials and Methods 

### 3.1. Chemistry

#### 3.1.1. Materials and General Methods

All the reagents were obtained commercially and used without further purification. Thiazolidine-2,4-dione **1** and (4-oxo-2-thioxo-3-thiazolidinyl)acetic acid **3** were purchased from Sigma-Aldrich. Purity of the compounds was checked by thin layer chromatography using silica-gel 60 F254-coated Al plates (Merck) and spots were observed under UV light. ^1^H NMR and ^13^C NMR spectra were recorded on a Bruker Avance spectrometer (600 and 150 MHz, respectively) and Varian VXR-400 spectrometer at 400 MHz and 101 MHz, respectively, at 298 K in CDCl_3_ or DMSO-*d*_6_ at a concentration of samples of 5–15 mmol, with TMS as internal reference for ^1^H and ^13^C NMR spectra. The signal assignments of compound **11**, **12**, **15**, and **17** were performed using 2D spectra (DQF-COSY, ^13^C–^1^H HSQC, and ^13^C–^1^H HMBC); the chemical shifts are expressed in ppm (δ scale) using DMSO and CDCl_3_ as an internal standard, and the coupling constants are expressed in Hz. The mass-spectral measurements were carried out by ESI method on micrOTOF-QII (Brucker Daltonics GmbH). Analytical HPLC was performed on a Shimadzu LC-20AD system using Kromasil-100-5-C18 (Akzo-Nobel) column, 4.6 × 250 mm, temperature 20 °C, UV detection, mobile phase A—0.2% HCOONH_4_), mobile phase B-MeCN, (pH 7.4), and fl-1ml/min.

#### 3.1.2. Synthesis of the Starting Materials

The synthetic routes used to synthesize starting materials are outlined in Figure 1, Figure 2 and Figure 3. The detailed description of the method and physico-chemical properties of **6a**, **8**, and **9** were described elsewhere [56,60].

**Ethyl 2,4-dioxothiazolidine-3-carboxylate (3)**. To a solution of thiazolidine-2,4-dione, (**1**) (11.7 g, 0.1 mol) and Et_3_N (28.0 mL, 0.2 mol) in dry toluene (100 mL), ethyl chloroformate (10.5 mL, 0.11 mol) was added in small portions, keeping the temperature at 3–5 °C. After stirring at room temperature for 3 h, water (250 mL) was added; then, the mixture was filtrated, the filtrate was diluted with EtOAc (100 mL), the organic layer was separated, and the aqueous part was re-extracted with EtOAc (100 mL). Organic extracts were combined, dried with Na_2_SO_4_, and evaporated in vacuo. The residue was purified by flash chromatography (100 g of silica gel) using CHCl_3_-hexane mixtures (10:1 to 1:0) as an eluent. The product was dried in vacuo at room temperature to give **3** (18.1 g, 96%) as colorless oil. ^1^H NMR (400 MHz, CDCl_3_) δ 4.38 (2H, q, *J* = 7.16), 4.00 (2H, s), 1.34 (3H, t, *J* = 6.82). ^13^C NMR (101 MHz, CDCl_3_) δ 167.65, 167.55, 147.49, 65.88, 33.93, 13.81. HRMS (EI): Calcd for C_6_H_7_NO_4_S [M + H]^+^ 190.0169. Found: *m/z* 190.0135.

***(Z)*-****5-(4-Chlorobenzylidene)thiazolidine-2,4-dione (4a)** [53,67,68,69,70,71,72]. A solution of 4-chlorobenzaldehyde (147.5 g. 1.05 mol), 2,4-thiazolidinedione (**1**) (117.2 g, 1.0 mol), 33% aqueous solution of methylamine (0.2 mL, 1.6 mmol) in acetic acid (500 mL) was refluxed for 6 h. The reaction mixture was cooled to room temperature and evaporated in vacuo. The residue was diluted with water (100 mL), and the resulted precipitate was filtered off, washed with cold water (100 mL), and dried under vacuum. The residue was dissolved in CHCl_3_ (200 mL), filtered, and then i-PrOH (200 mL) was added, the resulting suspension filtered, washed with cold i-PrOH (50 mL), and dried in vacuo to give **4a** (179.7 g, 75%) as colorless crystals. Filtrates were combined and evaporated in vacuo and the residue was purified by flash chromatography (100 g of silica gel) using CHCl_3_-hexane mixtures (10:1 to 1:0) as an eluent, to give **4a** (50.3 g, 21%) as colorless crystals. ^1^H NMR (400 MHz, DMSO-*d*_6_) δ 12.63 (1H, s, NH), 7.71 (1H, s, CH=C), 7.53 (4H, t, *J*
*=* 9.62,Ar), ^13^C NMR (101 MHz, DMSO-d6) δ 167.99, 167.59, 135.42, 132.30, 132.00(2C), 130.82, 129.74(2C), 124.66. HRMS (EI): Calcd for C_10_H_5_NO_4_S [M + H]^+^ 238.0169. Found: *m/z* 238.0135.

***(Z)*-****5-(3,4-bis(benzyloxy)benzylidene)thiazolidine-2,4-dione (4b)**. The same procedure as described for **4a** was carried out using 3,4-bis(benzyloxy)benzaldehyde (31.8 g, 0.1 mol) and 2,4-thiazolidinedione (**1)** (11.1 g, 0.095 mol) to give **4b** (37.8 g, 91%) as colorless crystals. ^1^H NMR (600 MHz, CDCl_3_) δ 9.20 (1H, s), 7.68 (1H, s), 7.43 (4H, t, *J* = 7.9), 7.36 (4H, t, *J* = 7.5), 7.30 (2H, t, *J* = 7.4), 7.03 (1H, dd, *J* = 8.4, 2.1), 6.99 (1H d, *J* = 2.1), 6.96 (1H, d, *J* = 8.4), 5.21 (2H, s), 5.19 (2H, s). ^13^C NMR (151 MHz, CDCl_3_) δ 167.39, 166.89, 151.19, 148.77, 136.38, 136.24, 134.31, 128.63, 128.61, 128.10, 128.08, 127.23, 127.18, 126.06, 125.32, 119.64, 115.76, 114.08, 71.20, 70.88. HRMS (EI): Calcd for C_24_H_19_NO_4_S [M + H]^+^ 418.1035. Found: *m/z* 418.1085.

***(Z)*-****5-(2,4-Dichlorobenzylidene)thiazolidine-2,4-dione (4c)** [69,70]. The same procedure as described for **4a** was carried out using 2,4-dichlorobenzaldehyde (7.8 g, 50 mmol) and 2,4-thiazolidinedione **1** (5.2 g, 45 mmol) to give **4c** (1.6 g, 85%) as colorless crystals. ^1^H NMR (400 MHz, *DMSO-d6*) δ 12.73 (1H, s, NH), 7.81 (1H, s, C=CH), 7.76 (1H, s, Ar), 7.55–7,52 (2H, m, Ar). ^13^C NMR (101 MHz, *DMSO-6)*δ: 167.72 (C^2^=O), 167.97 (C^4^=O), 135.88, 135.79, 130.52, 130.40, 128.72, 128.34, 125.97. HRMS (EI): Calcd for C_10_H_4_Cl_2_NO_2_S [M + H]^+^ 274.1233. Found: *m/z* 274.1199.

***(Z)*-****5-(2-Hydroxy-5-chlorobenzylidene)thiazolidine-2,4-dione (4d) [70]**. The same procedure as described for **4a** was carried out using 2-hydroxy-5-chlorobenzaldehyde (8.8 g, 50 mmol) and 2,4-thiazolidinedione **1** (5.2 g, 45 mmol) to give **4d** (10.7 g, 84%) as yellow crystals. ^1^H NMR (400 MHz, DMSO-d6): δ 12.40 (1H, s, NH), 10.78 (1H, s, OH), 7.68 (1H, s, C=CH), 7.29 (1H, dd, *J* = 8.50, 1.97, Ar), 7.21 (1H, d, *J* = 1.83, Ar), 6.94 (1H, dd, *J*
*=* 8.4, 2.1, Ar). ^13^C NMR (101 MHz, DMSO-*d*_6_) δ 168.17, 167.81, 156.40, 131.91, 127.74, 125.95, 124.22, 123.49, 122.06, 118.20. HRMS (EI): Calcd for C_10_H_6_ClNO_3_S [M + H]^+^ 255.9830. Found: *m/z* 255.9867.

***(Z)*-****5-(2-Hydroxy-3,5-dichlorobenzylidene)thiazolidine-2,4-dione (4e)**. The same procedure as described for **4a** was carried out using 2-hydroxy-3,5-dichlorobenzaldehyde (9.5 g, 50 mmol) and 2,4-thiazolidinedione (**1**) (5.2 g, 45 mmol) to give **4e** (12.3 g, 87%) as yellow crystals. ^1^H NMR (400 MHz, DMSO-d6): δ 12.50 (1H, s. NH), 10.73 (1H, s, OH), 7.86 (1H, s, C=CH), 7.57 (1H, d, *J* = 2.24, Ar), 7.18 (1H, d, *J*
*=* 2.1, Ar). ^13^C NMR (101 MHz, DMSO-*d*_6_) δ 167.49, 167.06, 151.37, 130.69, 126.23, 125.88, 125.34, 124.52, 123.86, 123.18. HRMS (EI): Calcd for C_10_H_5_Cl_2_NO_3_S [M + H]^+^ 289.9440. Found: *m*/*z* 289.9457.

**Ethyl *(Z)*-5-(4-chlorobenzylidene)-2,4-dioxothiazolidine-3-carboxylate (5a)** [45]. To a solution of *(Z)*-5-(4-chlorobenzylidene)thiazolidine-2,4-dione (**4****a**) (23.9 g, 0.1 mol) and Et_3_N (28.0 mL, 0.2 mol) in dry toluene (200 mL), ethyl chloroformate (10.5 mL, 0.11 mol) was added in small portions, keeping the temperature at 3–5 °C. After stirring at room temperature for 3 h, water (250 mL) was added; then, the mixture was filtrated, the filtrate diluted with EtOAc (100 mL), the organic layer was separated, and the aqueous part was re-extracted with EtOAc (100 mL). Organic extracts were combined, dried with Na_2_SO_4_, and evaporated in vacuo. The residue was purified by flash chromatography (100 g of silica gel) using CHCl_3_-hexane mixtures (10:1 to 1:0) as an eluent, to give **5a** (3.4g, 11%) as colorless crystals. ^1^H NMR (400 MHz, CDCl_3_): δ 1.43 (3H, t, *J* = 7.70), 4.50 (2H, q, *J* = 7.15), 7.43 (4H, q, *J* = 6.89), 7.85 (1H, s). ^13^C NMR (101 MHz CDCl_3_): δ 13.9, 66.0, 120.4, 129.6, 131.2, 131.4, 134.0, 137.2, 147.3, 162.4, 163.3. HRMS (EI): Calcd for C_13_H_10_ClNO_4_S [M + H]^+^ 312.0092. Found: *m/z* 312.0104.

#### 3.1.3. Synthesis of the Target Compounds

**Isobutyl (Z)-5-(4-chlorobenzylidene)-2,4-dioxothiazolidine-3-carboxylate (5b)**. The same procedure as described for **5a** was carried out using *(Z)*-5-(4-chlorobenzylidene)thiazolidine-2,4-dione (**4a**) (23.9 g, 0.1 mol) and isobutyl chloroformate (14.2 mL, 0.11 mol) to give **5b** (29.2 g, 86%) as colorless crystals. ^1^H NMR (400 MHz, DMSO-d6): δ 1.02 (6H, d, *J* = 6.72), 2.04–2.14 (1H, m), 4.22 (2H, d, *J* = 6.38), 7.43 (4H, q, *J* = 7.15), 7.87 (1H, s). ^13^C NMR (101 MHz, DMSO-d6) δ 18.76 (2C), 27.66, 75.60, 120.41, 129.66(2C), 131.27, 131.43(2C), 133.93, 137.18, 147.47, 162.44, 163.32. HRMS (EI): Calcd for C_15_H_14_ClNO_4_S [M + H]^+^ 340.0405. Found: *m/z* 340.0376.

**Allyl *(Z)*-5-(4-chlorobenzylidene)-2,4-dioxothiazolidine-3-carboxylate (5c)**. The same procedure as described for **5a** was carried out using *(Z)*-5-(4-chlorobenzylidene)thiazolidine-2,4-dione (**4a**) (23.9 g, 0.1 mol) and allyl chloroformate (11.6 mL, 0.11 mol) to give **5c** (27.1 g, 84%) as colorless crystals. ^1^H NMR (400 MHz, CDCl_3_) δ 4.91 (2H, d, *J* = 5.57), 5.37 (1H, d, *J* = 10.52), 5.15 (1H, d, *J* = 17.29), 5.94–6.04 (1H, m), 7.43 (1H, q, *J* = 7.06), 7.85 (1H, s). ^13^C NMR (101 MHz, CDCl_3_) δ 65.07. 115.48, 115.79, 124.92(2C), 125.06, 126.47, 126.71(2C), 129.33, 132.48, 142.42, 157.56, 158.51. HRMS (EI): Calcd for C_14_H_10_ClNO_4_S [M + H]^+^ 324.0092. Found: *m/z* 324.0062.

**Phenyl *(Z)*-5-(4-chlorobenzylidene)-2,4-dioxothiazolidine-3-carboxylate (5d)**. The same procedure as described for **5a** was carried out using *(Z)*-5-(4-chlorobenzylidene) thiazolidine-2,4-dione (**4a**) (23.9 g, 0.1 mol) and phenyl chloroformate (13.8 mL, 0.11 mol) to give **5d** (28.1 g, 78%) as colorless crystals. ^1^H NMR (400 MHz, CDCl_3_) δ 7.31–7.35 (3H, m), 7.43–7.50 (6H, m), 7.94 (1H, s). ^13^C NMR (101 MHz, CDCl_3_) δ 120.05. 120.85(2C), 127.15, 129.73(2C), 129.77(2C), 131.17, 131.51(2C), 134.48, 137.40, 145.94, 150.18, 162.18, 163.17. HRMS (EI): Calcd for C_17_H_10_ClNO_4_S [M + H]^+^ 360.0092. Found: *m/z* 360.0105.

**Benzyl *(Z)*-5-(4-chlorobenzylidene)-2,4-dioxothiazolidine-3-carboxylate (5e)**. The same procedure as described for **5a** was carried out using of *(Z)*-5-(4-chlorobenzylidene) thiazolidine-2,4-dione (**4a**) (23.9 g, 0.1 mol) and benzyl chloroformate (15.7 mL, 0.11 mol) to give **5e** (26.8 g, 72%) as colorless crystals. ^1^H NMR (400 MHz, CDCl_3_) δ 4.90 (2H, s), 7.29–7.35 (3H, m), 7.39–7.45 (6H, m), 7.83 (1H, s). ^13^C NMR (101 MHz, CDCl_3_) δ 45.36, 122.08, 128.32, 128.76(2C), 128.89(2C), 129.54(2C), 131.28(2C), 131.65, 132.47, 135.02, 136.64, 165.92, 167.28. HRMS (EI): Calcd for C_18_H_12_ClNO_4_S [M + H]^+^ 374.0248. Found: *m*/*z* 374.0288.

**5-(4-Chlorobenzylidene)-3-benzoythiazolidine-2,4-dione (7a)**. 5-(4-Chlorobenzylidene)thiazolidine-2,4-dione potassium salt^72^ (27.7 g, 0.1 mol) was suspended in dry acetone (200 mL) and benzoyl chloride (12.7 mL, 0.11 mol) was added at room temperature. The reaction mixture was stirred at 60 °C for 2 h and cooled to room temperature and filtered; then, acetone was evaporated in vacuo. The residue was diluted with water and CHCl_3_ (200 + 200 mL). Then, the organic layer was separated, dried by Na_2_SO_4_, and evaporated in vacuo. The residue was recrystallized from absolute EtOH to give **7a** (29.5 g, 86%) as colorless crystals. ^1^H NMR (600 MHz, CDCl_3_) δ 7.91 (2H, d, *J* = 7.8), 7.87 (1H, s), 7.69 (1H, t, *J* = 7.5), 7.52 (2H, t, *J* = 7.8), 7.49–7.42 (4H, m). ^13^C NMR (151 MHz, CDCl_3_) δ 166.12, 164.96, 164.33, 137.22, 135.60, 133.94, 131.44, 131.27, 130.78, 130.74, 129.68, 129.16, 121.20. HRMS (EI): Calcd for C_17_H_10_ClNO_3_S [M + H]^+^ 344.0143. Found: *m/z* 344.0188.

**(3,4-Dibenzyloxibenzylidene)-3-benzoythiazolidine-2,4-dione (7b)**. The same procedure as described for **7a** was carried out using *(Z)*-5-(3,4-dibenzyloxibenzylidene) thiazolidine-2,4-dione **4b** (41.7 g, 0.1 mol) and benzoyl chloride (12.7 mL, 0.11 mol). to give **7b** (40.6 g, 78%) as colourless crystals. ^1^H NMR (600 MHz, CDCl_3_) δ 7.90 (2H, d, *J* = 7.7), 7.78 (1H, s), 7.67 (1H, t, *J* = 7.4), 7.50 (2H, t, *J* = 7.7), 7.47–7.29 (10H, m), 7.07 (1H, dd, *J* = 8.4, 2.1), 7.03 (1H, d, *J* = 2.1), 7.00 (1H, d, *J* = 8.3), 5.24 (2H, s), 5.22 (2H, s). ^13^C NMR (151 MHz, CDCl_3_) δ 166.46, 165.53, 164.59, 151.54, 148.87, 136.41, 136.24, 135.47, 135.43, 130.93, 130.76, 129.09, 128.67, 128.65, 128.12, 128.10, 127.13, 127.12, 125.95, 125.61, 117.75, 115.96, 114.16, 71.27, 70.89. HRMS (EI): Calcd for C_31_H_23_ClNO_5_S [M + H]^+^ 522.370. Found: *m/z* 522.1402.

**Ethyl 2-(2,4-dioxothiazolidin-3-yl)acetate (8)** [60]. To a solution of 2,4-thiazolidinedione (1 g, 8.5 mmol) and ethyl bromoacetate (3.2 g, 17 mmol) in THF (25 mL) was added potassium carbonate (2.35 g, 17 mmol). The suspension was refluxed for 5 h; then, the solvent was evaporated under reduced pressure. The residue was washed with methanol to provide (2,4-dioxothiazolidin-3-yl)acetic acid ethyl ester (1.62 g, 93%) as white oil. ^1^H NMR (400 MHz, DMSO-d6) δ 4.32 (2H, s), 4.20 (2H, s), 4.29 (2H, s), 4.15–4.10 (2H, q, *J* = 7.1), 1.18 (3H, t, *J* = 7.1). ^13^C NMR (101 MHz, DMSO-d6) δ 172.20, 171.77, 167.14, 61.93, 42.16, 42.34, 34.43, 14.36. HRMS (EI): Calcd for C_7_H_9_NO_4_S [M + H]^+^ 204.0325. Found: *m/z* 204.0355.

**2-(2,4-Dioxothiazolidin-3-yl)acetic acid (9)** [60]. A solution of **8** (1.0 g, 5 mmol) in glacial acetic acid (25 mL) with addition of 2 mL 2N HCl was refluxed for 2 h. After evaporation to dryness in vacuo, the crude oil was washed with water and ethanol to provide pure (2,4-dioxothiazolidin-3-yl)acetic acid (0.82 g, 94%) as a white oil. ^1^H NMR (400 MHz, DMSO-*d*_6_): δ 13.30 (1H, s), 4.32 (2H, s), 4.20 (2H, s). ^13^C NMR (101 MHz, DMSO-*d*_6_) δ 171.90, 171.48, 168.11, 42.06, 34.04.

***(Z)-tert*-****Butyl-2-(5-(4-chlorobenzylidene)-2,4-dioxothiazolidin-3-yl)acetate (10a)**. The same procedure as described for **8** was carried out using *(Z)*-5-(4-Chlorobenzylidene) thiazolidine-2,4-dione **4a** (23.9 g, 0.1 mol) and *tert*-butyl bromoacetate (17.8 mL, 0.11 mol) to give **10a** (41.4 g, 78%) as colorless crystals. ^1^H NMR (600 MHz, CDCl_3_) δ 7.76 (1H, s), 7.43 (4H, t, *J* = 7.7), 7.36 (5H, t, *J* = 7.8), 7.30 (2H, dt, *J* = 7.8, 1.8), 7.00–6.95 (2H, m), 5.20 (4H, d, *J* = 7.1), 4.34 (2H, s), 1.45 (9H, s).^13^C NMR (101 MHz, DMSO-*d*_6_) δ 167.04, 165.98, 165.29, 135.97, 133.16, 132.32(2C), 132.07, 129.92(2C), 121.70, 82.98, 43.35, 27.98(3C). HRMS (EI): Calcd for C_16_H_16_ClNO_4_S [M + H]^+^ 354.0561. Found: *m*/*z* 354.0578.

***(Z)-tert*-****Butyl-2-(5-(3,4-bis(benzyloxy)benzylidene)-2,4-dioxothiazolidin-3-yl)acetate (10b)**. The same procedure as described for **8** was carried out. *(Z)*-5-(3,4-dibenzyloxibenzylidene)thiazolidine-2,4-dione (**4b**) (41.7 g, 0.1 mol) and *tert*-butyl bromoacetate (17.8 mL, 0.11 mol) were used to give **10b** (41.4 g, 78%) as colorless crystals. ^1^H NMR (600 MHz, CDCl_3_): δ 7.76 (1H, s), 7.43 (4H, t, *J* = 7.7), 7.36 (5H, t, *J* = 7.8), 7.30 (2H, dt, *J* = 7.8, 1.8), 7.00–6.95 (2H, m), 5.20 (4H, d, *J* = 7.1), 4.34 (2H, s), 1.45 (9H, s). ^13^C NMR (151 MHz, CDCl_3_): δ 167.55, 165.69, 165.22, 136.47, 136.45, 134.34, 128.63(2C), 128.60(2C), 128.04(2C), 127.13(3C), 127.09(2C), 126.25, 125.25, 118.46, 115.83, 114.15, 83.13, 71.20, 70.85, 42.77, 27.94(3C). HRMS (EI): Calcd for C_30_H_29_NO_6_S [M + H]^+^ 532.1788. Found: *m/z* 532.1774.

**Ethyl *(Z)*-2-(5-(4-chlorobenzylidene)-2,4-dioxothiazolidin-3-yl)acetate (11a) [57]**. To a solution of *(Z)*-5-(4-chlorobenzylidene)thiazolidine-2,4-dione (**4a**) (23.9 g, 0.1 mol) in THF (250 mL), ethyl bromoacetate (12.1 mL, 0.11 mol) and K_2_CO_3_ (27.5 g, 0.2 mol) were added. The mixture was refluxed for 5 h, and then cooled to room temperature, filtrated, and the filtrate evaporated under reduced pressure. The residue was dissolved in CHCl_3_/water (100 + 100 mL), the organic layer was separated, it was dried by Na_2_SO_4_, and it was evaporated in vacuo. The product was recrystallized from dry EtOH to give **11a** (25.3 g, 78%) as colorless crystals. ^1^H NMR (600 MHz, CDCl_3_) δ 7.90 (1H, s), 7.46 (2H, d, *J* = 8.4), 7.42 (2H, d, *J* = 8.4), 4.68 (2H, s), 4.22 (2H, q, *J* = 7.1), 1.27 (3H, t, *J* = 7.1). ^13^C NMR (151 MHz, CDCl_3_) δ 164.67, 164.34, 163.23, 137.60, 134.95, 131.55, 129.77, 119.39, 113.90, 44.82 42.07, 14.08. HRMS (EI): Calcd for C_14_H_12_ClNO_4_S [M + H]^+^ 326.0248. Found: *m/z* 326.0259.

**Ethyl-2-(5-(3,4-bisbenzyloxybenzylidene)-2,4-dioxothiazolidin-3-yl)acetate (11b)**. The procedure for **8** was carried out using *(Z)*-5-(3,4-dibenzyloxibenzylidene)thiazolidine-2,4-dione (**4b**) (41.7 g, 0.1 mol) and ethyl bromoacetate (12.1 mL, 0.11 mol) to give **11b** (34.7 g, 69%) as colorless crystals. ^1^H NMR (600 MHz, CDCl_3_) δ 7.77 (1H, s), 7.46–7.29 (10H, m), 7.05 (1H, dd, *J* = 8.4, 2.1), 7.02 (1H, d, *J* = 2.1), 6.97 (1H, d, *J* = 8.4), 5.22 (2H, s), 5.21 (2H, s), 4.43 (2H, s), 4.22 (2H, q, *J* = 7.1), 1.27 (3H, t, *J* = 7.1). ^13^C NMR (151 MHz, CDCl_3_) δ 167.55, 166.27, 165.63, 151.25, 148.84, 136.47, 136.35, 134.61, 128.66, 128.09, 127.16, 127.12, 126.23, 125.35, 118.36, 115.88, 114.20, 71.26, 70.91, 62.10, 42.07, 14.08. HRMS (EI): Calcd for C_28_H_25_NO_6_S [M + H]^+^ 504.1475. Found: *m/z* 504.1501.

***(Z)*-****2-(5-(4-chlorobenzylidene)-2,4-dioxothiazolidin-3-yl)acetic acid (12a)** [82].

**Method 1. (From 10a)** A solution of ethyl *(Z)*-2-(5-(4-chlorobenzylidene)-2,4-dioxothiazolidin-3-yl)acetate (**10a**) (16.2 g, 50 mmol) in AcOH (150 mL) with addition of 2 N HCl (50 mL) was refluxed for 4 h. After evaporation to dryness in vacuo, the crude oil was washed with water and then ethanol to provide pure **12a** (12.7 g, 86%) as colorless crystals.

**Method 2. (From 11a)** To the solution of *tert*-butyl *(Z)*-2-(5-(4-chlorobenzylidene)-2,4-dioxothiazolidin-3-yl)acetate (**11a**) (17.6 g, 50 mmol) in CH_2_Cl_2_ (250 mL), was added TFA (25 mL, 0.3 mol); then, reaction mixture was stirred at room temperature for 8 h. The solvent was removed under reduced pressure and the residue was dried under vacuum overnight to obtain **12a** (14.4 g, 97%) as colorless crystals.

^1^H NMR (600 MHz, CDCl_3_) δ 7.90 (1H, s), 7.46 (2H, d, *J* = 8.4), 7.42 (2H, d, *J* = 8.4), 4.68 (2H, s). ^13^C NMR (151 MHz, CDCl_3_) δ 164.67, 164.34, 163.23, 137.60, 134.95, 131.55, 129.77, 119.39, 113.90, 44.82. HRMS (EI): Calcd for C_12_H_8_ClNO_4_S [M + H]^+^ 297.9935. Found: *m/z* 297.9991.

***(Z)*-****2-(5-(3,4-Bis(phenylmethoxy)phenyl]methylene]-)-2,4-dioxothiazolidin-3-yl)acetic acid (12b)** [12].

**(From 10b)** The same procedure as for **12a** (Method 1) was carried out. *(Z)*-*tert*-Butyl 2-(5-(3,4-bisbenzyloxybenzylidene)-2,4-dioxothiazolidin-3-yl)acetate **10b** (26.5 g, 50 mmol) was used in the synthesis to give **12b** (13.7 g, 58%) as colorless crystals.

**(From 11b)** The same procedure as for **12a** (Method 2) was carried out using **11b** (5.4 g, 10 mmol) and glacial acetic acid (17 mL) with 2 N HCl (4.2 mL) to give 1.6 g of **12b** (34% yield).

**(From 9)** According to the method described for **4a**, 3,4-bis(benzyloxy)benzaldehyde (33.4 g, 0.105 mol) and (2,4-dioxothiazolidin-3-yl)acetic acid (**9**) (17.5 g, 0.1 mol) were used to give **12b** (32.8g, 69%), as colorless crystals.

^1^H NMR (600 MHz, CDCl_3_) δ 9.20 (1H, s), 7.68 (1H, s), 7.43 (4H, t, *J* = 7.9), 7.36 (4H, t, *J* = 7.5), 7.30 (2H, t, *J* = 7.4), 7.03 (1H, dd, *J* = 8.4, 2.1), 6.99 (1H, d, *J* = 2.1), 6.96 (1H, d, *J* = 8.4), 5.21 (2H, s), 5.19 (2H, s). ^13^C NMR (151 MHz, CDCl_3_) δ 167.39, 166.89, 151.19, 148.77, 136.38, 136.24, 134.31, 128.63, 128.61, 128.10, 128.08, 127.23, 127.18, 126.06, 125.32, 119.64, 115.76, 114.08, 71.20, 70.88. HRMS (EI): Calcd for C_26_H_21_NO_6_S [M + H]^+^ 476.162. Found: *m/z* 476.1145.

***(Z)*-****2-(5-(2,4-Dichlorobenzylidene)-2,4-dioxothiazolidin-3-yl)acetic acid****(12c)**. According to the method described for **4a**, 2,4-dichlorobenzaldehyde (17.5 g, 0.105 mol) and (2,4-dioxothiazolidin-3-yl)acetic acid (**9**) (17.5 g, 0.1 mol) were used to give **12c** (28.8 g, 87%) as colorless crystals. ^1^H NMR (DMSO-*d*_6_): 13.46 (1H, br. s), 8.03 (1H, s), 7.88 (1H, s), 7.65 (2H, s), 4.41 (2H, s). ^13^C NMR (400 MHz, DMSO-*d*_6_) δ: 168.26, 166.85, 164.97, 136.38, 135.91, 130.67, 130.43, 128.84, 128.29, 125.49, 42.88. HRMS (EI): Calcd for C_12_H_7_Cl_2_NO_4_S [M + H]^+^ 333.1585. Found: *m/z* 333.1608.

***(Z)*-****2-(5-(2-Hydroxy-5-chlorobenzylidene)-2,4-dioxothiazolidin-3-yl)acetic acid****(12d)**. According to the method described for **4a**, 2-hydroxy-5-chlorobenzaldehyde(13.3 g, 0.105 mol) and (2,4-dioxothiazolidin-3-yl)acetic acid (**9**) (17.5 g, 0.1 mol) were used to give **12d** (21.6 g, 69%) as yellow crystals. ^1^H NMR (400 MHz, DMSO-*d*_6_) δ 13.30 (1H, s), 10.97 (1H, s), 8.01 (1H, s), 7.35 (1H, dd, *J* = 8.81, 2.35), 7.30 (1H, d, *J* = 1.97), 6.97 (1H, d, *J* = 8.81), 4.35 (2H, s). ^13^C NMR (101 MHz, DMSO-*d*_6_) δ 168.41, 167.26, 165.46, 156.47, 132.51, 128.51, 128.32, 123.58, 121.75, 121.51, 118.36, 42.72. HRMS (EI): Calcd for C_12_H_8_ClNO_5_S [M + H]^+^ 313.9884. Found: *m/z* 313.9906.

***(Z)*-****2-(5-(2-Hydroxy-3,5-dichlorobenzylidene)-2,4-dioxothiazolidin-3-yl)acetic acid (12e)**. According to the method described for **4a**, 2-hydroxy-3,5-dichlorobenzaldehyde (20.0 g, 0.105 mol) and (2,4-dioxothiazolidin-3-yl)acetic acid (**9**) (17.5 g, 0.1 mol) were used to give **12e** (24.7 g, 71%), as yellow crystals. ^1^H NMR (400 MHz, DMSO-*d*_6_) δ 8.00 (1H, s), 7.61 (1H, d, *J* = 2.22), 7.28 (1H, d, *J* = 2.14), 4.35 (2H, s). ^13^C NMR (101 MHz, DMSO-*d*_6_) δ 168.35, 167.10, 165.25, 152.24, 131.57, 128.42, 127.19, 124.60, 123.95, 123.82, 123.37, 42.83. HRMS (EI): Calcd for C_12_H_7_C_l2_NO_5_S [M + H]^+^ 347.9495. Found: *m/z* 347.9496.

***(Z)*-****5-(3,4-Bis(benzyloxy)benzylidene)-2-thioxothiazolidin-4-one (15b)**. According to the method described for **4a**, 3,4-bis(benzyloxy)benzaldehyde (31.8 g, 0.1 mol) and 2-thioxothiazolidin-4-one (**13**) (12.6 g, 0.095 mol) were used to give **15b** (38.2 g, 93%), as yellow crystals. ^1^H NMR (600 MHz, CDCl_3_) δ 9.45 (1H, s), 7.51 (1H, s), 7.46–7.43 (4H, m), 7.42 (2H, s), 7.37 (6H, td, *J* = 7.5, 4.1), 7.31 (3H, t, *J* = 7.3), 7.03 (1H, dd, *J* = 8.4, 2.1), 6.98–6.96 (2H, m), 5.23 (3H, s), 5.21 (3H, s). ^13^C NMR (151 MHz, CDCl_3_) δ 167.39, 166.89, 151.19, 148.77, 136.38, 136.24, 134.31, 128.63, 128.61, 128.10, 128.08, 127.23, 127.18, 126.06, 125.32, 119.64, 115.76, 114.08, 71.20, 70.88. HRMS (EI): Calcd for C_26_H_21_NO_5_S_2_ [M + H]^+^ 492.0934. Found: *m/z* 492.0958.

***(Z)*-****2-(5-(3,4-Bis(benzyloxy)benzylidene)-4-oxo-2-thioxothiazolidin-3-yl)acetic acid (17b)** [12].

**Method 1**. **(From 14)** To a solution of 2-(4-oxo-2-thioxothiazolidin-3-yl)acetic acid (19 g, 0.1 mol) in AcOH (250 mL) were added 3,4-bis(benzyloxy)benzaldehyde (33.4 g, 0.105 mol) and 33% aqueous methylamine (0.2 mL, 1.6 mmol). Reaction mixture was refluxed for 6 h then poured into H_2_O to obtain a crude solid. The suspension was filtered and the precipitate recrystallized from *i*-PrOH to give **17b** (35.4 g, 72%), as yellow crystals.

**Method 2**. **(****From 15b)** To a solution of *(Z)*-5-(3,4-bis(benzyloxy)benzylidene)-2-thioxothiazolidin-4-one (**15b**) (4.17 g, 0.01 mol) in THF (50 mL), *tert*-butyl bromoacetate (1.78 mL, 0.011 mol) and K_2_CO_3_ (2.75 g, 0.02 mol) were added. The mixture was refluxed for 5 h, cooled to room temperature, filtrated, and then the solvent was evaporated under reduced pressure. The residue was redissolved in a mixture of CHCl_3_ (50 mL) and water (50 mL); organic layer was separated, dried by Na_2_SO_4_, and evaporated in vacuo and the residue used without additional purification for the hydrolysis with trifluoroacetic acid (25 mL) in CH_2_Cl_2_ (100 mL). The product was recrystallized in dry EtOH to give **17b** (3.2 g, 65%) as colorless crystals.

^1^H NMR (400 MHz, DMSO-*d*) δ: 13.7 (1H, br. s., OH), 7.52–7.44 (11H, m), 7.37–7.32 (3H, m), 5.19 (2H, s), 5.18 (2H, s). ^13^C NMR (101 MHz, DMSO-*d*) δ 195.89 (C=S), 169.80 (C=O), 151.05, 148.66, 137.20, 132.41, 128.90, 128.40, 128.36, 128.04, 127.97, 126.28, 125.46, 123.05, 114.57, 70.46, 70.34. HRMS (EI): Calcd for C_26_H_21_NO_5_S_2_ [M + H]^+^ 492.5835. Found: *m/z* 492.5887.

***(Z)*-5-(2-Hydroxy-3,5-dichlorobenzylidene)-4-oxo-2-thioxo-3- thiazolidine acetic acid (17e)**. The same procedure as described for **17b** (Method 1) was carried out using 2-(4-oxo-2-thioxothiazolidin-3-yl)acetic acid (19 g, 0.1 mol) and 2-hydroxy-3,5-dichlorobenzaldehyde (20.0 g, 0.105 mol) to give **17e** (20.7 g, 57%), as yellow crystals. ^1^H NMR (600 MHz, DMSO-*d*_6_) δ 13.45 (1H, s), 10.97 (1H, s), 7.90 (1H, s), 7.69 (1H, d, *J* = 2.5), 7.36 (1H, d, *J* = 2.5), 4.73 (2H, s). ^13^C NMR (600 MHz, DMSO-*d*_6_) δ: 193.28 (C^2^=S), 167.21(C^4^=O), 166.19, 131.57, 128.05, 127.79, 124.15, 124.05, 123.29, 45.01. HRMS (EI): Calcd for C_12_H_7_Cl_2_NO_3_S [M + H]^+^ 347.9317. Found: *m/z* 347.9474.

### 3.2. Biochemistry and Microbiology

#### 3.2.1. MIC Determination

The minimal inhibitory concentrations (MICs) of the compounds obtained were determined using the microdilution method in RPMI 1640 medium buffered to pH 7.0 with MOPS [74]. The test compounds were evaluated against human pathogens *Candida* spp and dermatophytes. Fluconazole was used as the reference antifungal drug. Compounds were dissolved in DMSO and serially diluted in nutrient medium (concentration range 0.125–128 mg/L). The inoculum suspension was added to each well and incubated at 35 °C. Microdilution plates were visually inspected after 24–48 h and 96 h of incubation for growth of yeast and filamentous fungi, respectively. MIC was defined as the minimum inhibitory concentration of the test compound which resulted in total inhibition of the fungal growth. All susceptibility testing was performed in triplicate. Clinical and reference strains of *Candida* spp. were: *C. parapsilosis* ATCC 22019, *C. albicans* ATCC 24433, *C. tropicalis* 3019, *C. kefyr* 77, *C. famata* 312, *C. guilliermondii* 355, *C. parapsilosis* 58N, *C. krusei* 432M, and *C. glabrata* 61L. Clinical and reference strains of filamentous fungi were: *A. niger* 37a, *Aspergillus fumigatus* ATCC 46645, *Microsporum canis B-200*, and *Trichophyton rubrum* 2002. 

*Saccharomyces cerevisiae*, strain BY4741 (*MATa his3Δ1 leu2Δ0 met15Δ0 ura3Δ0)* was inoculated into YPD medium (1% Yeast extract (*w*/*v*), 2% Peptone (*w*/*v*), 2% Glucose (*w*/*v*)) supplemented with 2-fold differing concentrations of Mycosidine (stock solution 20 mg/mL) in DMSO) and a constant concentration (0.05% *v*/*v*) of dimethylsulfoxide (Sigma) in 96 well plates (200 µL final volume), at an initial OD_600_ of 0.05. Cell growth was judged after 24 h of growth at 30 °C, with the MIC being the lowest concentration of drug where no growth was observed. Statistical processing of the data obtained, shown in the tables, was carried out by traditional methods of mathematical statistics using standard software for IBM PC: Microsoft Office Excel 2013. Differences in mean values and between groups were assessed using Student’s *t*-test, set at the level of *p* < 0.05.

#### 3.2.2. Growth Assays and Cell Death Detection

*S. cerevisiae*, strain BY4742 (*MATa his3Δ1 leu2Δ0 lys2Δ0 ura3Δ0)*, and its derivative deletion strains, lacking the *PDR5* and *HXT3* genes (obtained from the Yeast Genome Deletion Collection) were grown in YPD medium to logarithmic phase (OD_600_ = 0.2–0.7). They were then either spotted onto solid YPD medium with different concentrations of Mycosidine at a starting OD_600_ of 1 with 10-fold serial dilutions (growth assays) or incubated in distilled water supplemented with different concentrations of Mycosidine for different periods of time (OD_600_ = 0.05), after which the treated cells were spotted onto YPD medium in stepwise 10-fold serial dilutions.

#### 3.2.3. Flow Cytometric Analysis of Changes in GFP-Tagged Protein Abundance

Derivatives of the BY4741 strain obtained in [75], harboring C-terminal GFP tags on proteins of interest, were used to detect changes of protein level in response to drug treatment. Genes encoding these proteins were modified in the genome at their 3′-terminus, which corresponds to the C-terminus of the protein [75], i.e., the regulatory regions of the genes were identical to those of the wild-type proteins. In general, this type of tagging rarely interferes with protein function or regulation [77]. Cells bearing specific GFP fusions were pinned onto solid YPD medium (2% agar) (*w*/*v*), grown overnight to form small colonies, and then inoculated into liquid medium containing either the vehicle (DMSO) or Mycosidine and incubated for 6 h. One hour prior to the end of incubation, propidium iodide was added to the incubating cells at a concentration of (1 mcg/mL). After this, the control sample and drug-treated sample were analyzed on a Cytoflex S flow cytometer (Beckman Coulter) equipped with a 96-well sampler. GFP fluorescence was assayed using the 488 nm laser and FITC filter, whereas propidium iodide staining was assayed with the 532 nm laser and the PE filter. 

Raw data were analyzed using CytExpert software (Beckman Coulter), and graphs were obtained by exporting data to MS Excel. Heatmaps were obtained using MS Excel and they depict the log_2_ transformed ratio between the GFP fluorescence in the experimental sample divided by the control sample. 

#### 3.2.4. Sample Preparation for Cell Wall Investigation

*C. parapsilosis* 22,019 cells and *C. albicans* 10,231 cells were used for experiments. The topography of yeast cells was obtained in physiological saline at room temperature. A daily exposition of *Candida* suspension (10^3^ CFU/mL) with drugs was performed in 5 mL petri dishes (GBO, Kremsmünster, Austria) modified with a layer of SYLGARD™ 527 A&B Silicone Dielectric Gel (Dow Chemical, Midland, MI, USA) to increase cell-surface adhesion and prevent sample drifting and reattachment. The surface of a petri dish was covered with the silicone gel with a ratio of components A and B in 1: 1 (by mass fraction) and kept in a laboratory drying cabinet Ulab UT-4686 (ULAB, Dhaka, Bangladesh) for an hour at 60 °C; then, the dishes were washed with distilled water. Concentration of cells in the suspension in physiological saline (PanEco, RF) was determined using hemocytometer (MiniMed, RF) and added to the modified petri dish. Dry drugs were dissolved in DMSO 99.7% (Sigma-Aldrich, St. Louis, MO, USA) and added to cells in physiological saline with final concentrations 16, 160 µg/mL of Mycosidine and 32, 320 µg/mL of **17**. After incubation of cells for 24 h, the dishes were washed and filled with saline.

#### 3.2.5. Scanning Ion-Conductance Microscopy (SICM)

SICM experiments were performed at room temperature by using SICM by ICAPPIC (ICAPPIC Ltd., London, UK) and borosilicate glass capillaries (Sutter Instruments, Novato, CA, USA). Nanopipettes were made from capillaries with an external diameter of 1.2 mm and an internal diameter of 0.69 mm on a P-2000 puller (Sutter Instruments, Novato, CA, USA). Topography was measured in saline [78] on cells immobilized on a layer of silicone dielectric gel. SICM measurements were performed using hopping mode protocol [79]. Set-point parameter of ion current feedback control was 0.2%, providing a minimal influence of nanopipette probe on the sample. The nanopipette radius ranged from 20 to 35 nm to achieve a nanoscale resolution of topography imaging. The radius of the nanopipette and the mechanical characteristics were calculated using formulas reported in previous studies [80]. Cell images were recorded on areas of 10 × 10 µm at 74 nm resolution and on areas of 1.5 × 1.5 µm for 21 nm resolution. The image dimensions were 256 × 256 pixels. For each concentration point, three to five large images were obtained for a total of 5 to 10 cells. Image processing was performed using the “SICMImageViewer” software.

## 4. Conclusions

Novel derivatives of Mycosidine—3,5-substituted thiazolidine-2,4-diones were synthesized in excellent yield using Knoevenagel synthesis and alkylation or acylation of the imide nitrogen. Furthermore, 5-Arylidene-2,4-thiazolidinediones and 2-thioxo analogs containing halogen and hydroxy groups or a di(benzyloxy) group in the 5-benzylidene moiety were evaluated for their in vitro antifungal activity. Some of them exhibited strong activity against filamentous fungi (*Trichophyton rubrum* and *Microsporum canis*) and *Candida* spp. yeasts. The compounds were characterized by chromatographic and spectrometric methods. The compounds exhibited both fungistatic and fungicidal activity and resulted in the emergence of morphological changes in the cell wall of the *Candida* yeast. Using limited proteomic screening and the analysis of toxicity in mutants, we show that Mycosidine activity is dependent on glucose transport. This suggests that this first-in-class antifungal and its derivatives have a novel mechanism of action, which deserves further study. 

## Data Availability

Data is contained within the article and Appendix A.

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
