# Peer review of "Antifungal Thiazolidines: Synthesis and Biological Evaluation of Mycosidine Congeners"

_pharmaceuticals, 2022, doi:10.3390/ph15050563_

Round 1

Reviewer 1 Report

The research presents the synthesis and testing of the antifungal activity of some thiazolidine compounds. Some of the compounds in the article have been reported in the literature and their synthesis has been reproduced: 4a, 4c, 4d, 5a, 11a, 12a and 17b.  The authors mentioned the bibliographic references accordingly.

The manuscript has several errors:- lines 75, 76 (PPR -g, PPR - gamma is correct), - line 89 - TNf-a, TNH-alpha is correct- spaces between words: line 133, 89, 244, 334, 341 , 345, 346, 486, 510, 828.

  • mycosidine is written with either M or m, so it needs to be standardized
  • the design of chemical formulas and reaction schemes needs to be improved. Also, the way in which the compounds are numbered and the conditions should be reconsidered. For example, in Scheme 4 it appears denoted by e some reaction conditions, but also appears a compound denoted by 12e. I propose the numbering of the reaction conditions, in each scheme with i, ii, iii, iv, etc., in order to not create confusion and to make the article more easier to read.
  • in schemes 4 and 5, I recommend replacing Ar-COH with Ar-CHO
  • row 210 what does ease mean?
  •  line 211, why byproduct?
  • line 239, replace hypoglycemic with anti-hyperglycemic
  • Candida albicans should be written in italics throughout the article. The same for in vivo and in vitro
  • line 196, 5-arylidene
  • Scheme 4, line 277, 5-arylidene instead of benzylidene. Similar in Figure 5. It is a substituted benzylidene residue, it is preferable to be called arylidene
  • Scheme 5, line 283, requires another one parenthesis before acetic
  • line 288, to replace on with in
  • line 321, to replace for with with
  • In Table 3, dichlor (di-Cl) would be more correct instead of Cl2. For example, in compound 4e, 2-OH-3,5-di-Cl. Or 2-OH-3,5-(Cl)2.  Similar for 12c, 12 e, 17e
  • line 648 - correct ofethyl with of ethyl
  • The bibliographic references are a lot for an experimental article but, in general, they are a old 

Author Response

Dear reviewer,

Thank you very much for your corrections, including the smallest ones. All the suggestions were noted and the manuscript changed accordingly.

Reviewer 2 Report

Antifungal thiazolidines. Synthesis and biological evaluation
of mycosidine congeners is original and well conceptualized manuscript. But, there are a few shortcomings that need to be corrected in order for the manuscript to be published.  So, comments\suggestions about the manuscript are as follows :

in vitro, in vivo and Candida albicans are written italic. Correct this, especially in Abstract and Introduction.

Fig. 1-3 are they original authors original work or they are downloaded? Please specify. 

The tables title should be written above not below the table. Also table 3 are very confused. There are no explanation below the table what is R1 and R2. Further, it is not emphasized that the presented values are  MICs, and there are missing the units of measurement (μg/ml) in the table and in the table title. Also, in table 3 authors need to be specify which exact fungal isolate they used for MIC determination. So, please add ATCC number or code if it’s not ATCC strain used. Also, add that code numbers in the table title. 

lines 348 and 352 spp. its write regular not italic. Authors have the same mistake in Conclusion, so please check whole manuscript, for this kind of mistakes italic or not italic. 

Section 3.2.1. - add codes/ ATCC numbers for every fungal species, not only for Candida species.

Add section statistical analysis in Material and methods.

Overall, this manuscript have very interesting results but it lack more discussion. In every paragraph of the results there are only one or two sentence of other researchers results. If there is no data in the literature with which your data can be compared you must emphasize.Therefore, the discussion section needs to be expanded. 

Author Response

«in vitro, in vivo and Candida albicans are written italic. Correct this, especially in Abstract and Introduction.»
>>Corrected

«Fig. 1-3 are they original authors original work or they are downloaded? Please specify. 
>>Except for Mycosidine, all the compounds in the Introduction section aren't related to the authors of the current manuscript.»

«The tables title should be written above not below the table. Also table 3 are very confused. There are no explanation below the table what is R1 and R2. Further, it is not emphasized that the presented values are  MICs, and there are missing the units of measurement (μg/ml) in the table and in the table title. Also, in table 3 authors need to be specify which exact fungal isolate they used for MIC determination. So, please add ATCC number or code if it’s not ATCC strain used. Also, add that code numbers in the table title. »
>>Corrected. R1 and R2 are shown in the formula above the Table 3. The strains are described more specific now.

«lines 348 and 352 spp. its write regular not italic. Authors have the same mistake in Conclusion, so please check whole manuscript, for this kind of mistakes italic or not italic.»
>>Corrected

«Section 3.2.1. - add codes/ ATCC numbers for every fungal species, not only for Candida species.»
>>Added

«Add section statistical analysis in Material and methods.»
>>Added

Reviewer 3 Report

Peer review report on “Antifungal thiazolidines. Synthesis and biological evaluation of mycosidine congeners”.

Manuscript ID: Pharmaceuticals-1678818

This manuscript describes the preparation and the exploration of the antifungal bioactivity of analogues of Mycosidine which has been in commercial use for some years. It is generally well-written, and the experimental work has been effectively designed and clearly described.

Some comments:

Please label all spectra in the supplementary information with page number, eg., S1, S2, S3 etc, compound number, type of experiment, solvent, and instrument frequency. Renumber structures so that they are consistent with the manuscript text.

Mycosidine is a brand name for a drug. Please capitalize all references.

Please renumber the Figures consecutively. They skip from 3 to 6 to 8. Where are Figures 4, 5, and 7?

Please format the Tables, Figures and Schemes consistently as shown in Scheme 1, with the heading in bold style.

All the aldehyde reagents in the Schemes should be written as e.g., Ar-CHO, not Ar-COH.

Table 1:            In the heading, please state the operating frequency of the NMR instrument used, and in the table itself the multiplicities and coupling constants J in Hz.

Table 2:            State operating frequency of the NMR instrument in the heading. The 13C data do not match the NMR spectrum in the supplementary information, or the data in the experimental section for 5a. Please correct.

Table 3:            This requires a heading for the table columns stating compound, then MIC (µg/mL), as shown in Table 4.

Figure 8:           Label the B graph.

Section 2.6:       Clarify why you chose to use 17b instead of 12e for the effects on cell wall integrity when 17b is practically inactive against the fungi and 12e is quite potent.

Line 115:          Other, not others.

Lines 194 and 196:        … -thiazolidine-, not “-tiazolidine-“.  Hyphenate compound on line 196.

Lines 226 and 231:        Clarify that by (Fig. S1) and (Fig. S2, S3) and (Fig. S3) you are talking about Supplementary Information 3.  Otherwise, they don’t appear to reference anything.

Line 244:          “…. slightchangesand….”. Separate words.

Line 270:          “…. (benzoyloxy)…”  should be …(benzyloxy)…

Line 350:          Replace “Low MIC values….” with “Potent MIC values…..” and state what these values were.

Line 371:          Figure 8A not 6A.

Line 382:          Figure 8B not 6B.

Lines 581, 591, 615, 661, and 732:         Italicize m/z

Author Response

(The authors gave the same response as above.)

Reviewer 4 Report

The article by Levshin et al. is part of a series of research on finding new antifungal agents, with a new mechanism of action.

This is an extensive study of organic synthesis and spectral characterization, as well as an investigation of the antifungal potential of newly synthesized compounds, and the authors must be congratulated for their intensive research activity.

The current study is intended to be a molecular modelling and drug design study aimed the obtaining thiazolidine-2,4-dione derivatives with an improved antifungal effect. I do not understand why the authors refer to the substance that they call Mycosdine, about which very little information is found in the literature.

As I found in the literature, “3-methoxycarbonyl-5- (4-chlorobenzylidene) thiazolidine-2,4-dione is known as a compound that is part of the drug“ Mycosidine ”with antimicrobial activity (patent SU No. 1417436, published April 27, 1996 ”.

I also did not find mycosidine in the ATC Index.

That is why I consider it is wrong to relate the antifungal activity of these compounds to that of Mycosidine. The comparison with (Z) -5-(4-chlorobenzylidene)-3-methoxycarbonyl-1,3-thiazolidine-2,4-dione would have been correct. 

That is why I believe that the article needs to be revised, and the title should be changed as well.

why I believe that I have a few more comments for the authors:

It is recommended that Latin words (in vivo, in vitro, via, in vacuo), including fungal strains, be written in italics.

For millilitres it is better to write mL.

The authors state in the Introduction that azoles inhibit ergosterols synthesis (row 32). Ergosterol should be used correctly.

There are many words combined in the article, such as confirmedusing (line 133), slightchangesand (line 244), beassociated (line 334).

I would recommend the authors in describing the syntheses to use room temperature everywhere instead of rt.

I find the article interesting, but it needs to be revised before it can be published.

Author Response

«The current study is intended to be a molecular modelling and drug design study aimed the obtaining thiazolidine-2,4-dione derivatives with an improved antifungal effect. I do not understand why the authors refer to the substance that they call Mycosdine, about which very little information is found in the literature.
As I found in the literature, “3-methoxycarbonyl-5- (4-chlorobenzylidene) thiazolidine-2,4-dione is known as a compound that is part of the drug“ Mycosidine ”with antimicrobial activity (patent SU No. 1417436, published April 27, 1996 ”.
I also did not find mycosidine in the ATC Index.
That is why I consider it is wrong to relate the antifungal activity of these compounds to that of Mycosidine. The comparison with (Z) -5-(4-chlorobenzylidene)-3-methoxycarbonyl-1,3-thiazolidine-2,4-dione would have been correct.»

>> (Z)-5-(4-chlorobenzylidene)-3-methoxycarbonyl-1,3-thiazolidine-2,4-dione is the Mycosidine's active ingredient. It is used in Russia only, so it's not found in ATC or anywhere else.

«It is recommended that Latin words (in vivo, in vitro, via, in vacuo), including fungal strains, be written in italics.
For millilitres it is better to write mL.
The authors state in the Introduction that azoles inhibit ergosterols synthesis (row 32). Ergosterol should be used correctly.
There are many words combined in the article, such as confirmedusing (line 133), slightchangesand (line 244), beassociated (line 334).
I would recommend the authors in describing the syntheses to use room temperature everywhere instead of rt.»
>> Corrected

Round 2

Reviewer 1 Report

I agree with the publication in this form.

Author Response

Dear reviewer,

thank you very much for your time.

Reviewer 4 Report

I appreciate the changes made by the authors and consider that the scientific quality of the article has increased. The synthesis of original molecules is very difficult to achieve and therefore the authors should be congratulated.

One minor correction:

Line 108
The chemical name must be corrected using chloro

I still think it would have been more correct for the authors to give a coded name Mycosidine and not use this name as a reference, as long as it is not an INN name.

In this way, confusion with commercial pharmaceutical product Mycosidine, which is not just the active substance ((Z)-5-(4-chlorobenzylidene)-3-methoxycarbonyl- 1,3-thiazolidine-2,4-dione), would have been avoided.

If there are no similar opinions, I agree with the publication of the article, after the suggested minor correction is made.

Author Response

Dear reviewer,

thank you very much for your suggestions and your time.